# Combined Regional Approach of Talimogene laherparepvec and Radiotherapy in the Treatment of Advanced Melanoma

**DOI:** 10.3390/cancers16111951

**Published:** 2024-05-21

**Authors:** Andrew Tam, Colton Ladbury, Ari Kassardjian, Badri Modi, Heather McGee, Laleh Melstrom, Kim Margolin, Yan Xing, Arya Amini

**Affiliations:** 1Department of Radiation Oncology, City of Hope Comprehensive Cancer Center, 1500 E Duarte Rd., Duarte, CA 91010, USA; atam@coh.org (A.T.);; 2Department of Dermatology, City of Hope Comprehensive Cancer Center, 1500 E Duarte Rd., Duarte, CA 91010, USA; 3Department of Surgery, City of Hope Comprehensive Cancer Center, 1500 E Duarte Rd., Duarte, CA 91010, USA; 4St. John’s Cancer Institute, 2121 Santa Monica Blvd., Santa Monica, CA 90404, USA; 5Department of Medical Oncology and Therapeutics Research, City of Hope Comprehensive Cancer Center, 1500 E Duarte Rd, Duarte, CA 91010, USA

**Keywords:** melanoma, immunotherapy, TVEC, radiotherapy

## Abstract

**Simple Summary:**

Talimogene laherparepvec (TVEC), a modified virus, is a local injected therapy that has been demonstrated to be an effective treatment in patients with advanced melanoma. Recent reports have suggested that radiation treatment (RT) may activate the immune system when combined with TVEC to further improve both local and systemic responses. However, studies on the combined effect of TVEC and RT remain limited. Here, we reviewed twenty melanoma patients who had received TVEC and RT (fourteen patients to the same region and six patients to different regions of the body) and found a benefit in progression-free survival (PFS) and time to distance metastasis (DM) with a combined regional approach of TVEC and RT. These results suggest that the regional approach of using these two treatment modalities may confer a therapeutic benefit.

**Abstract:**

Talimogene laherparepvec (TVEC) is a genetically modified oncolytic herpes simplex virus (HSV-1) that is used for the intralesional treatment of advanced or metastatic melanoma. Given that TVEC produces the granulocyte–macrophage colony-stimulating factor (GM-CSF), recent reports have suggested that radiation treatment (RT) given in conjunction with TVEC may provide synergistic immune activation at the site, and possibly systemically. However, studies on combining RT with TVEC remain limited. We conducted a retrospective review of melanoma patients from a single cancer center who received TVEC and RT in the same region of the body and compared them to patients who received TVEC with RT at another site (other than the site of TVEC injection). Between January 2015 and September 2022, we identified twenty patients who were treated with TVEC and RT; fourteen patients received TVEC and RT in the same region, and six had treatments in separate regions. Regions were determined at the time of analysis and were based on anatomic sites (such as arm, leg, torso, etc.). Kaplan–Meier analysis of progression-free survival (PFS), analyses of time to distant metastasis (DM), overall survival (OS), and locoregional control (LRC), and the corresponding log-rank test were performed. With a median follow-up of 10.5 months [mos] (range 1.0–58.7 mos), we found an improvement in PFS with TVEC and RT in the same region compared to different regions, which were 6.4 mos (95% CI, 2.4–NR mos) and 2.8 mos (95% CI, 0.7–4.4 mos), respectively; *p* = 0.005. There was also a significant improvement in DM when TVEC and RT were used in the same region compared to different regions: 13.8 mos (95% CI, 4.6–NR mos) and 2.8 mos (95% CI, 0.7–4.4 mos), respectively (*p* = 0.001). However, we found no difference in overall survival (OS) between patients who had TVEC and RT in the same region (19.0 mos, 95% confidence interval [CI], 4.1–not reached [NR] mos) and those who received treatments in different regions (18.5 mos, 95% CI, 1.0–NR mos); *p* = 0.366. There was no statistically significant improvement in locoregional control (LRC) in patients who had TVEC and RT in the same region was 26.0 mos (95% CI, 6.4–26.0 mos) compared to patients who received TVEC and RT in different regions (4.4 mos) (95% CI, 0.7–NR mos) (*p* = 0.115). No grade 3 or higher toxicities were documented in either group. Overall, there were improvements in PFS and DM when TVEC and RT were delivered to the same region of the body compared to when they were used in different regions. However, we did not find a significant difference in locoregional recurrence or OS. Future studies are needed to assess the sequence and timing of combining RT and TVEC to potentially enhance the immune response both locally and distantly.

## 1. Introduction

Talimogene laherparepvec (TVEC) is an oncolytic herpes simplex virus type-1-derived immunotherapy delivered by intralesional injection for the treatment of advanced or metastatic melanoma [1]. It was first demonstrated to have therapeutic benefits in the Oncovex Pivotal Trial in Melanoma (OPTiM) trial, a phase III randomized controlled trial that compared TVEC to granulocyte–macrophage colony-stimulating factor (GM-CSF) in patients with unresectable stage IIIB to IV melanoma. The trial found improvement in the durable response rate with TVEC compared to the control (19.0% vs. 1.4%, unadjusted odds ratio [OR] 16.6, *p* < 0.0001) [1,2]. More notably, there was an improvement in the median overall survival (OS) (23.3 months [mos] vs. 18.9 months, unstratified hazard ratio [HR] 0.79, *p* = 0.0494) [2].

The underlying mechanism of TVEC is thought to be due to the genetically modified virus’s ability to preferentially replicate in tumor cells leading to tumor lysis and the release of tumor-associated antigens [3]. These antigens and other damage-associated molecular patterns (DAMPs) induce both a local inflammatory response and activate dendritic cells to induce CD8+ T cell-mediated adaptive immunity [3]. Given TVEC’s ability to elicit a local antitumor response, recent efforts have focused on combining TVEC with other therapies to generate a systemic response. A phase II randomized trial comparing TVEC and ipilimumab vs. ipilimumab alone found an improvement in the objective response rates of both injected (39% vs. 18%, OR 2.9, *p* = 0.002) and visceral lesions (52% vs. 23%) with combined TVEC + ipilimumab [4]. However, a phase III randomized controlled trial comparing TVEC and pembrolizumab vs. placebo pembrolizumab did not find any difference in progression-free survival (PFS) (HR 0.86; *p* = 0.13) or OS (HR 0.96, *p* = 0.74) [4].

Despite the results from this phase III trial with pembrolizumab, combining TVEC with other therapies, such as radiation treatment (RT), still holds promise in the treatment of melanoma. RT induces double-strand DNA breaks which lead to immunogenic cell death and can serve as an “in situ vaccine” [5]. Given the similarities in the immune response to viruses and the immune response to radiation [6], there is a strong rationale for combining these two treatments. Clinically, the abscopal effect of treatment with local RT for melanoma leading to a regression of metastases has been reported in a handful of case reports and small series [7,8]. A phase II randomized trial (NCT02819843) on combining RT with TVEC vs. TVEC alone in solid tumors was closed prematurely due to slow accrual and no observed overall responses; although, a potential signal was seen in cutaneous metastases [9].

Per current practice guidelines, the role of RT in the management of melanoma remains limited and it should be considered for certain circumstances based on multidisciplinary discussions [10,11,12]. However, there is currently limited guidance on conjunctive treatment with TVEC and RT. In this study, we compared patients who received TVEC and RT in the same region of the body to those whose RT region did not also include the site of the TVEC injection.

## 2. Materials and Methods

We conducted a retrospective review of the medical charts of patients with stage IIIB to stage IV melanoma who received TVEC and RT from a single comprehensive cancer center between January 2015 and September 2022. Demographics including date of birth, sex, and race, and ethnicity were collected. Clinical data such as oncologic history (stage and molecular profile), treatments details (date of TVEC injections, location[s] of TVEC administration, dates of RT, RT dose and fractionation, and RT site[s]), oncologic outcomes (locoregional recurrence, distant metastases, and date of death), and adverse events graded under the terms of the Common Terminology Criteria of Adverse Events (CTCAE) were recorded. The follow-up period is defined as the date of completion of both TVEC and RT to the most recent oncologic follow-up visit. The study cohort was divided into two groups based on sites treated with TVEC and RT by defining the following as regions of body: left arm, right arm, left leg, right leg, torso, scalp and neck, and brain. We considered the brain as a separate region from the scalp given the brain’s unique immune environment that differs from that of other organs [13]. Patients who had TVEC injections and RT in the same region of the body were grouped into the “same region” arm and those who did not have treatment to the same region were sorted into the “different regions” arm.

OS was defined as being from the date of completion of both treatments to the date of death or last follow-up. Progression-free survival (PFS) was defined as from the date of completion of both treatments to the date of the first event, with event defined as either development of locoregional recurrence, distant metastasis, or death. Time to distant metastasis (DM) was time from completion of both treatments to the date of the development of any new site of metastasis or progression of preexisting metastatic site(s). Distant metastasis was assessed using all available data including all imaging (computed tomography [CT], magnetic resonance imaging [MRI], and positron emission tomography [PET]). Lastly, time to locoregional control (LRC) was the time from completion of both treatments to the date of development of recurrence at the previously treated (either TVEC or RT) site or region.

Demographics were presented either with median values and ranges or percentages. OS, PFS, DM, and LRC were compared using Kaplan–Meier analysis and the corresponding log-rank test. Skin toxicity was also recorded. Analyses were performed using, version 4.4.0 (Vienna, Austria). Statistical significance was defined as *p* < 0.05. The study was approved by our institutional review board under IRB 23192.

## 3. Results

A total of twenty patients were identified who received both TVEC and RT during the specified timeframe; fourteen patients were treated with TVEC and RT in the same region of the body and six received TVEC and RT to separate regions. Of the patients who had received both treatments to the same region, the following regions were treated: left leg (*n* = 5, 36%), right leg (*n* = 5, 36%), torso (*n* = 1, 7%), left arm (*n* = 1, 7%), right arm (*n* = 1, 7%), and scalp (*n* = 1, 7%). Of the patients that received the treatments to different regions, four different sites were treated with TVEC (two [33%] scalp, two [33%] torso, one [17%] left leg, and one [17%] right leg) and three different sites were treated with RT (four [67%] brain, one [17%] spine, and one [17%] right leg).

The patient demographics are summarized in Table 1. All of the characteristics were well balanced between the two groups. The median ages were 75.5 years (range 44 to 90 years) and 67.3 years (range 59 to 79 years) in the same region group and different regions, respectively. Sex was evenly represented in both groups with 50% of both sexes included in both arms. The majority of patients identified as White and presented with stage III disease. The majority of patients with stage III disease had undergone a sentinel lymph node biopsy (SLNBx) and lymph node dissection (LND) as part of their staging. Somatic testing showed that the tumor of one (7%) patient in the ”same region” cohort and one (17%) patient in the “different regions” cohort had activating *BRAF* mutations. The majority of tumors from both groups (50% in the “same region” arm and 67% in the “different region” arm) showed programmed death ligand-1 (PD-L1) expression ≥1%. None of the patients received TVEC and RT concurrently; 64% of patients had TVEC first followed by RT in the “same region” group, compared to 50% in the “different regions” group. The median time interval between the two treatments was 6.6 mos (range 0.6 to 39.7 mos) in the “same region” arm and 10.5 months (range 1.2 to 46.7 mos) in the “different regions” arm. All patients (100%) in both groups had received an immune checkpoint blockade during their course of treatment. Four (29%) patients in the “same region” arm and one (17%) patient in the “different regions” arm were treated with chemotherapy. Regarding RT, in the “same region” arm, ten (71%) patients were treated with stereotactic body radiation treatment (SBRT) to the same region as their TVEC injection (dose ranged 27 to 40 Gy/3–5 fractions [fx]), while six (43%) patients were treated with non-SBRT doses (dose ranged 48 to 70 Gy/20–35 fx). In the “different regions” arm, five (83%) patients and one (17%) patient had received RT to areas not treated by TVEC injections with SBRT (14.8–40 Gy/3–5 fx) and hypofractionated (48 Gy/20 fx) doses, respectively.

Within the median follow-up time of 10.5 mos (range 1.0–58.7 mos), four (28.6%) patients and three (50.0%) patients had died. The median OS was 19.0 mos among patients who had TVEC and RT in the “same region” (95% confidence interval [CI], 4.1–not reached [NR] mos), compared to 18.5 mos among those who the received treatments in “different regions” (95% CI, 1.0–NR mos); there was no significant difference in OS between the two groups with *p* = 0.366 (Figure 1A). There was a significant difference in PFS between the “same region” arm (6.4 mos, 95% CI, 2.4–NR mos) and the “different regions” arm (2.8 mos, 95% CI, 0.7–4.4 mos); *p* = 0.005 (Figure 1B). The time to DM was 13.8 mos (95% CI, 4.6–NR mos) with TVEC and RT in the “same region” and 2.8 mos (95% CI, 0.7–4.4 mos) in “different regions” (*p* = 0.001) (Figure 1C). LRC was 26.0 mos (95% CI, 6.4–26.0 mos) and 4.4 mos (95% CI, 0.7–NR mos) for patients who received TVEC and RT in the “same region” vs. “different regions”, respectively (*p* = 0.115) (Figure 1D). There were no unexpected toxicities from either treatment modality.

## 4. Discussion

In this study of patients with advanced melanoma who received both TVEC and RT, we found that patients who received treatments to the same region appeared to have longer PFS and time to DM compared to patients who had treatments to different regions of the body. These findings suggest a potential systemic effect from combined TVEC and RT to the same site. However, this contrasts with the preliminary findings from a phase II randomized trial that found no observed responses in distant metastases with TVEC and RT (0%) when compared to TVEC alone (13%) [9]. One possible explanation for these discordant findings may be the cancer types involved as our study focused only on melanoma patients while the phase II trial included Merkel cell carcinoma and other solid tumors with skin metastases. Another explanation may be attributed to RT dose and fractionation. In the phase II trial, all patients received hypofractionated RT [14], while in our study, 71% of patients in the same region arm received SBRT. SBRT, characterized by high-dose radiation given in five or fewer fractionations, has long been thought to stimulate the anti-tumor immune response in ways that are not observed with conventional or hypofractionated RT [15]. Preclinical studies have demonstrated that radiation induces “immunogenic cell death”, promotes proinflammatory modifiers of the tumor microenvironment, and engages antitumor T cells [16,17,18]. This immunostimulatory effect is dose- and fractionation-dependent [19,20]. Clinically, trials on other cancer types have found a benefit of SBRT on distant metastatic control. For example, a phase II trial (NCT02239900) for ipilimumab with concurrent or sequential SBRT to metastatic lesions in the liver or lung found a 26% overall response rate in terms of nonirradiated (distal) tumor volume [21]. However, data from randomized controlled trials on the role of SBRT in melanoma remain limited, with some retrospective data suggesting improved regional and distant controls with SBRT when compared to conventional or hypofractionated RT [22]. Nevertheless, future studies are needed to better understand the potential abscopal response of combining SBRT with TVEC in melanoma patients.

Despite our results showing an improvement in DM with both TVEC and RT, this did not contribute to a benefit in OS. The lack of OS is likely attributed to the advanced disease stage among this cohort. TVEC is currently approved for patients with stage IIIB, IIIC, or IV cancers only [2,23]. It is important to note that, in the initial analysis of the OPTiM trial, there was no benefit to OS with TVEC when compared to the control (23.3 mos vs. 18.9 mos; HR 0.79, 95% CI 0.62 to 1.00 mos; *p* = 0.051), however, a statistical significance was noted (*p* = 0.049) in the final analysis. Our findings provide further evidence to suggest that distant metastatic control does not necessarily confer a benefit to OS. Our study also did not demonstrate a significant difference in LRC between the two arms despite there being a nominal difference in the median LRC (26.0 mos vs. 4.4 mos; *p* = 0.23). Given the retrospective design of this study and the small sample size, we are limited in commenting on the lack of benefits to OS or LRC from the improved distant metastatic control. Further studies are needed to better understand the relationship between the development of DM and OS.

To our knowledge, this was the first study that evaluated the possible impact of the choice of irradiated sites in relation to TVEC administration for melanoma. In this study, given the difficulty in ascertaining the exact location of the lesions that were treated with TVEC and radiation through our chart review, patients were grouped based on regions of the body. Interestingly, we observed a benefit to DM despite our focus on regional treatments. This suggests a possible systemic immune-stimulatory effect that might not necessitate an overlap of the treatment fields. In fact, given the radiosensitivity of immune cells and the acute effect of RT on immune cell death [24,25], non-superimposed treatment may potentially lead to a more robust immunogenicity. Our understanding of the biological underpinning of the abscopal effect is limited. Preclinical studies have reported that the combination of RT with immunotherapy improves cross-priming of T cells by enhancing the function of dendritic cells [17,26]. This, in turn, induces a systemic immunity effect against metastases which is mediated by T cells [17,27]. However, some preclinical data have also suggested that radiation induces immunosuppressive signals [28]. Nevertheless, there is a dearth of studies on the immune effect on regional treatments. Current trials, such as the phase II trial on melanoma and a phase IB/II trial on sarcoma [9,29], are focusing on delivery of RT and TVEC to the same site(s). Given our findings and the effect of radiation in inducing apoptosis of T cells, future studies may consider a regional approach considering that the delivery of RT and TVEC to the exact same site at the same time may suppress the T cells that are supposed to be activated by TVEC, especially if RT is delivered directly following TVEC.

The sequence of treatment RT and TVEC may have an important role in immunogenicity. Some preclinical studies have reported that the timing of these treatments matters and the effect is dependent on the type of immunotherapy [30,31,32]. Clinical data on the sequence of treatment are lacking. In this study, twelve patients received TVEC initially followed by RT while eight patients had RT first and then TVEC. Due to the small number of patients in this cohort, a sub-analysis was not performed. There are several limitations to our study, including its retrospective nature and small sample size. Even though demographics were well balanced between the two groups, there was a higher percentage of patients who received RT for brain metastases in the “different regions” group than in the “same region” group (14.3% [*n* = 2] in the “same region” arm vs. 83.3% [*n* = 5] in the “different regions” arm). As brain metastases are associated with poor prognosis, one may argue that patients in the “different regions” cohort have more advanced disease. However, interestingly, the “different regions” arm did not have poorer OS in this study. Other potential biases such as time-related bias, especially considering the nominal difference in the time interval (although not significant) between TVEC and RT of the two groups, may have contributed to our findings. Despite these limitations, the findings of this study provide important insights into the potential role of a regional approach to TVEC and RT in stimulating systemic immune responses. Future study designs incorporating TVEC and RT as an immunomodulator should also incorporate biomarkers to better predict treatment response and potentially select those who are more likely to progress locally versus systemically in guiding the appropriate local and/or systemic treatment modality [33].

## 5. Conclusions

In this retrospective study, patients who received TVEC and RT in the same region of the body had improvements in PFS and DM when compared to patients who received treatments in different regions of the body. However, we did not find a significant difference in LRC or OS. This suggests that a combined therapeutic approach of RT and TVEC to the same region could be a promising treatment strategy for melanoma. Larger trials are needed to better understand the findings from our study.

## Figures and Tables

**Figure 1 cancers-16-01951-f001:**
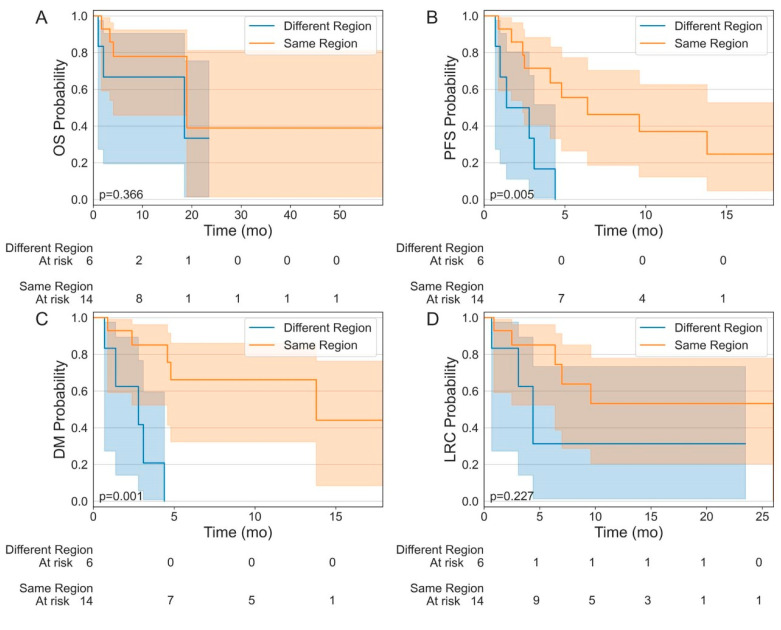
Kaplan–Meier estimates of (**A**) OS, (**B**) PFS, (**C**), DM, and (**D**) LRC of patients who had received TVEC and RT in the same region compared to those who had these treatments in different regions of the body.

**Table 1 cancers-16-01951-t001:** Patient Demographics.

	TVEC + RT to Same Region (*n* = 14)	TVEC + RT to Different Regions (*n* = 6)	*p* Value
**Age**, median (range)	75.5 (44 to 90) years	67.3 (59 to 79) years	0.099
**Sex**, *n* (%)			
Female	7 (50.0)	3 (50.0)	1.000
Male	7 (50.0)	3 (50.0)	
**Race**, *n* (%)			
Asian/Pacific Islander	1 (7.1)	0 (0.0)	0.679
Black/African American	1 (7.1)	0 (0.0)	
White	11 (78.6)	6 (100.0)	
Other/Unknown	1 (7.1)	0 (0.0)	
**Ethnicity**, *n* (%)			
Hispanic/Latino	6 (42.9)	2 (33.3)	0.692
Non-Hispanic/Latino	7 (50.0)	4 (66.7)	
Unknown	1 (7.1)	0 (0.0)	
**Stage**, *n* (%)			
Stage III	10 (71.4)	4 (66.7)	0.831
Stage IV	4 (28.6)	2 (33.3)	
**Had SLNBx as part of staging?**, *n* (%)			
Yes	[Stage III] 10 (100.0) [Stage IV] 2 (50.0)	[Stage III] 3 (75.0) [Stage IV] 2 (100.0)	0.101 0.221
**Had CLND for staging?**, *n* (%)			
Yes	[Stage III] 8 (80.0) [Stage IV] 0 (0.0)	[Stage III] 4 (100.0) [Stage IV] 1 (50.0)	0.334 0.121
**BRAF-mutation**, *n* (%)			
Positive	1 (7.1)	1 (16.7)	0.281
Negative	10 (71.4)	2 (33.3)	
Unknown	3 (21.4)	3 (50.0)	
**PD-L1 expression ≥1**, *n* (%)			
Positive	7 (50.0)	4 (66.7)	0.339
Negative	4 (28.6)	0 (0.0)	
Unknown	3 (21.4)	2 (33.3)	
**Treatment sequence**, *n* (%)			
TVEC first	9 (64.3)	3 (50.0)	0.550
RT first	5 (35.7)	3 (50.0)	
**Time Interval between TVEC + RT**, median (range)	6.6 (0.6 to 39.7) months	10.5 (1.2 to 46.7) months	0.458
**Regions treated by TVEC + RT**	left leg: 5 (35.7), right leg: 5 (35.7), torso: 1 (7.1), left arm: 1 (7.1), right arm: 1 (7.1), scalp: 1 (7.1)	n/a	n/a
**RT regimen**			
RT dose and fractionation	[to the region treated by TVEC + RT] 6 (42.9) [Conventional/Hypofractionated; 48–70 Gy/20–35 fx] 10 (71.4) [SBRT, 27–40 Gy/3–5 fx]	[to the regions not treated by TVEC] 1 (16.7) [Hypofractionated; 48 Gy/20 fx] 5 (83.3) [SBRT, 14.8–40 Gy/3–5 fx]	n/a
**Other treatments**			
Immunotherapy	14 (100.0)	6 (100.0)	1.000
Chemotherapy	4 (28.6)	1 (16.7)	0.573
Other (i.e., BRAF inhibitors and CMP-001)	5 (35.7)	4 (66.7)	0.202

CLND, complete lymph node dissection; n/a, not applicable; PD-L1, programmed death-ligand 1; RT, radiation treatment; SBRT, stereotactic body radiation treatment (high dose RT in five or fewer fractions); SLNBx, sentinel lymph node biopsy; TVEC, talimogene laherparepvec.

## Data Availability

The raw data supporting the conclusions of this article will be made available by the authors on request.

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
