# Peer review of "Combined Regional Approach of Talimogene laherparepvec and Radiotherapy in the Treatment of Advanced Melanoma"

_cancers, 2024, doi:10.3390/cancers16111951_

Round 1
Reviewer 1 Report
Comments and Suggestions for Authors
Interesting case series on a less explored field of melanoma management. After addressing few points, the article can become more comprehensive. For instance:
1. The study's focus on melanoma therapy using TVEC prompts the necessity to clearly specify patient stages, particularly as TVEC is approved for stages 3B/3c-IVa. Emphasizing the inclusion criteria based on these stages ensures the relevance and applicability of the findings to the intended patient population. Please specify.
2. I would suggest to include the number of patients who underwent sentinel lymph node biopsy (SNLB) and completion lymph node dissection (CLND), as these procedures can significantly influence disease staging and subsequent treatment decisions, particularly in stage 3 patients.
3. The observed difference in progression-free survival (PFS) between study cohorts warrants consideration of potential biases, such as lead time bias. The fixed time interval between TVEC and radiotherapy administration raises questions about whether the observed differences in PFS (4 months) could be attributed to this bias, highlighting the need for careful interpretation of the results.
4. The absence of data on radiation therapy (RT) doses in one of the study groups represents a notable limitation. Including this information is crucial for a comprehensive analysis of treatment outcomes and for elucidating any potential dose-response relationships.
5. How is the finding of differences in locoregional control (LRC) between the 2 study cohorts not significant (24 months vs 4 months)?
6. I would suggest to integrate the recommendations from the EADO guidelines (https://doi.org/10.1016/j.ejca.2022.04.018) which provide a valuable framework for the current best practices in melanoma management. Additionally, discussing the role of potential biomarkers in predicting treatment response, as highlighted in recent literature (DOI: 10.1080/14737159.2024.2314574), underscores the importance of ongoing research efforts in refining therapeutic strategies and improving patient outcomes.
Comments on the Quality of English LanguageGood.
Reviewer 2 Report
Comments and Suggestions for Authors
This retrospective study included 20 melanoma patients from a single cancer center, 14 patients received TVEC and RT in the same region of the body, and 6 had treatments in separate regions (other than the site of TVEC injection). The study is well designed and adequately performed and presented.
Minor comments:
1. In the Introduction it could be of interest to comment on the relevance of the multidisciplinary management of melanoma patients, referring to the following paper: Wouters MW, Michielin O, Bastiaannet E, Beishon M, Catalano O, Del Marmol V, Delgado-Bolton R, Dendale R, Trill MD, Ferrari A, Forsea AM, Kreckel H, Lövey J, Luyten G, Massi D, Mohr P, Oberst S, Pereira P, Prata JPP, Rutkowski P, Saarto T, Sheth S, Spurrier-Bernard G, Vuoristo MS, Costa A, Naredi P. ECCO essential requirements for quality cancer care: Melanoma. Crit Rev Oncol Hematol. 2018 Feb;122:164-178. doi: 10.1016/j.critrevonc.2017.12.020. Epub 2018 Jan 2. PMID: 29458785.
2. Line 37, check the phrase "There was no statistically significant improvement in , locoregional control (LRC) in patients"
